# Radiation Interception, Conversion and Partitioning Efficiency in Potato Landraces: How Far Are We from the Optimum?

**DOI:** 10.3390/plants9060787

**Published:** 2020-06-23

**Authors:** Cecilia Silva-Díaz, David A. Ramírez, Javier Rinza, Johan Ninanya, Hildo Loayza, René Gómez, Noelle L. Anglin, Raúl Eyzaguirre, Roberto Quiroz

**Affiliations:** 1International Potato Center (CIP), Headquarters P.O. Box 1558, Lima 12, Peru; silvadiaz.cc@gmail.com (C.S.-D.); j.rinza@cgiar.org (J.R.); j.ninanya@cgiar.org (J.N.); h.loayza@cgiar.org (H.L.); r.gomez@cgiar.org (R.G.); n.anglin@cgiar.org (N.L.A.); r.eyzaguirre@cgiar.org (R.E.); 2Water Resources Doctoral Program, Universidad Nacional Agraria La Molina (UNALM), Av. La Molina s/n, Lima 12, Peru; 3CATIE—Centro Agronómico Tropical de Investigación y Enseñanza, Cartago Turrialba 30501, Costa Rica; roberto.quiroz@catie.ac.cr

**Keywords:** mini core collection, diffuse radiation, tuberization precocity, senescence delay, remote sensing, crop modeling

## Abstract

Crop efficiencies associated with intercepted radiation, conversion into biomass and allocation to edible organs are essential for yield improvement strategies that would enhance genetic properties to maximize carbon gain without increasing crop inputs. The production of 20 potato landraces—never studied before—was analyzed for radiation interception ($\varepsilon_{i}$), conversion ($\varepsilon_{c}$) and partitioning ($\varepsilon_{p}$) efficiencies. Additionally, other physiological traits related to senescence delay (normalized difference vegetation index (NDVI)${}_{slp}$), tuberization precocity (tu), photosynthetic performance and dry tuber yield per plant (TY) were also assessed. Vegetation reflectance was remotely acquired and the efficiencies estimated through a process-based model parameterized by a time-series of airborne imageries. The combination of $\varepsilon_{i}$ and $\varepsilon_{c}$, closely associated with an early tuber maturity and a NDVI${}_{slp}$ explained 39% of the variability grouping the most productive genotypes. TY was closely correlated to senescence delay (r${}_{Pearson}$ = 0.74), indicating the usefulness of remote sensing methods for potato yield diversity characterization. About 89% of TY was explained by the first three principal components, associated mainly to tu, $\varepsilon_{c}$ and $\varepsilon_{i}$, respectively. When comparing potato with other major crops, its $\varepsilon_{p}$ is very close to the theoretical maximum. These findings suggest that there is room for improving $\varepsilon_{i}$ and $\varepsilon_{c}$ to enhance potato production.

## 1. Introduction

The predicted population of nine billion people by 2050 will require an increase in food production by at least 70%, under unfavorable environmental conditions [1]. The selection of improved genetic material by breeding programs could cope with present and future climatic challenges, and thus, deemed to be an adaptation action to face climate change [2]. A scope for future improvement considers that the physiological bases, together with genetic engineering efforts [3] could help increasing the potential yield of crops. These bases are regulated by genetically determined properties, intrinsic of each variety [4], and the available radiation energy, which depends on the site and year [5]. A change in any of these variables, would affect yield proportionally [6], e.g., crop growth could be analyzed in terms of the amount of radiation energy intercepted by the leaf and the efficiency of its use [5]. Notwithstanding, accounting for yield variation in terms of crop growth and development is complex, since additional external factors can also influence plant physiological processes, their interrelations as well as their dependence on the plant genotypic effect, which are difficult to measure under field conditions [7].

Potato has been highly recommended by the Food and Agriculture Organization of the United Nations (FAO) as an important food security crop, due to its widely adaptive range, great yield potential and high nutritional value [8]. There are more than 3500 native potatoes, mostly found in the Andes [9]. However, analysis for Andean landraces and hybrids are lacking in the literature, and therefore, a current understanding of potato physiology and modeling is needed for yield improvement and prediction [7], especially for the Andean region where climate change is affecting traditional farming practices and where potato is a staple food. The incorporation of remotely sensed data in models with different temporal resolution and levels of complexity can improve yield prediction in potato [10].

Genetic yield potential depends on three efficiencies: radiation interception ($\varepsilon_{i}$), conversion ($\varepsilon_{c}$) and partitioning ($\varepsilon_{p}$) [5]. $\varepsilon_{i}$ is affected by the speed of canopy development and closure, its longevity, size and architecture [4], from which canopy cover provides a good estimate of the fraction of photosynthetic active radiation (PAR) intercepted by the foliage [11,12]. The efficiency of converting the intercepted radiation into biomass ($\varepsilon_{c}$) is determined by the photosynthesis and respiration rate [4], and formalized as radiation use efficiency (RUE—[13,14]). RUE is directly linked to physiological process, which defines the ability of crops to grow and produce harvestable yields [15], it varies among and within species, and is influenced by the environment [16]. In potatoes, RUE tends to be stable throughout its growing period [17], and according to Quiroz et al. [10], the remotely sensed data are contingent on an appropriate RUE estimation. The proportion of biomass partitioned to the harvested plant organ ($\varepsilon_{p}$) is determined by the harvest index; in the case of potatoes, there is a favorable distribution of assimilates to tubers [18].

Referring to complementary traits, photosynthesis is a positive metric of crop growth, closely correlated to PAR canopy interception [19]. It is possible to increase its duration by delaying leaf senescence, which offers an opportunity for increasing the total amount of carbon fixed by a crop [10,20]. The delay of senescence that occurs in some potato varieties is defined as the slope of the pattern of greenness loss [21], calculated from chlorophyll concentration surrogate measurements [22], it has shown strong positive correlation with yield in potato [23,24]. Therefore, it is a desired trait in breeding programs because it is generally associated with an extension of photosynthetic activity [25]. Senescence delay (normalized difference vegetation index (NDVI)${}_{slp}$) is related to precocity or earliness (tuber initiation delay) which can be simulated in potato and sweetpotato as the thermal time at maximum tuber growth rate (tu) [24,26]. Thus, plants with low precocity show a delay in senescence, which extends the time of carbon assimilation and allocation to the tubers, ultimately obtaining higher yield. The Genebank from the International Potato Center (CIP) preserves the greatest diversity of potato cultivars, landraces and wild relatives worldwide [27]. Based on a combination of molecular marker data and curator’s knowledge to choose diverse sets with little genetic redundancy, the potato mini core subset is the most representative re-selection of CIP Genebank’s germplasm diversity [28]. In this paper, 20 genotypes belonging to this subset were assessed by the first time in an environment with high diffuse radiation, which promotes the highest reported RUE for potato [10]. It was hypothesized that this environmental condition would promote the potential expression of efficiencies allowing us to know the response of genetically diverse potatoes in comparison with other crops. This paper aims: (i) to analyze the relation between radiation interception, conversion and partitioning efficiencies, and plant traits related to senescence, precocity, photosynthetic performance and tuber yield, and (ii) to characterize the diversity of efficiency responses in relation to other crops.

## 2. Results

### 2.1. Relationship among Efficiencies and Physiological and Yield Traits

The analysis of variance for an augmented block design, applied to analyze the effect of accessions on the response of assessed variables, revealed significant differences among accessions (augmented) for dry tuber yield per plant (TY), $\varepsilon_{i}$, $\varepsilon_{c}$ and net photosynthesis rate (*A*), and within the improved varieties (checks) only for TY and $\varepsilon_{c}$ (Table 1). The average TY values ranged from 115 ± 12.6 to 665 ± 63.9 g plant${}^{-1}$, in which the highest value corresponded to CIP 703520 (*S. tuberosum subsp. andigenum*) (Figure 1A). Average values of TY for genotypes grouped by ploidy were 225.3 ± 26.1, 268.0 ± 29.7 and 372.5 ± 38.1 g plant${}^{-1}$ for 2×, 3× and 4× ploidy, respectively. Since 70% of plants belonging to the accession CIP 702650 (*S. ajuanhuiri*) did not produce aboveground biomass, it was not possible to have statistical repetitions for physiological analyses, and thus this accession was not considered for estimating the means presented in Figure 1.

Referring to the physiological traits, the adjusted values of *A* and the carbon isotopic discrimination in leaves ($\Delta_{leaf}$), as well as the senescence delay and precocity proxies NDVI${}_{slp}$ and tu, ranged between 17–26.6 μmol CO${}_{2}$ m${}^{-2}$ s${}^{-1}$, 21–23‰, −0.006 to −0.002 and 356.3–556.3 °C day${}^{-1}$, respectively (Table 2). The significantly higher and lower values of *A* were obtained in CIP 704406 (*S. tuberosum subsp. andigenum*) and CIP 706845 (*S. stenotomum subsp. stenotomum*), respectively. The highest NDVI${}_{slp}$ values were recorded by CIP 703520, CIP 702853 (*S. tuberosum subsp. andigenum*) and CIP 704406, which also had the lowest tu values. From these traits, tu and *A* were negatively and positively related with TY, respectively, and NDVI${}_{slp}$ presented the highest correlation (Table 3).

Concerning radiation interception, the cumulative PAR incident calculated until harvest dates (121 and 141 days after planting–dap), were 570 and 700 MJ m${}^{-2}$, respectively. Among accessions, CIP 703506 (*S. phureja*) and CIP 706845 (*S. Stenotomum subs. stenotomum*)—both harvested at 121 dap—showed the highest (69.3%) and lowest (15.4%) $\varepsilon_{i}$ values, respectively (Figure 1B). RUE, used to analyze resource capture and crop biomass accumulation, showed values between 1.0 and 3.6 g MJ${}^{-1}$, corresponding to a maximum $\varepsilon_{c}$ = 6.4%, observed in CIP 704406. The harvest index, descriptive of $\varepsilon_{p}$, ranged between 0.6 and 0.9, where CIP 705068 (*S. tuberosum subsp. tuberosum*) and CIP 704057 (*S. tuberosum subsp. andigenum*) had the highest (87%) and lowest (61%)–but not statistically different–$\varepsilon_{p}$ values (Figure 1D). Unlike $\varepsilon_{p}$, $\varepsilon_{i}$ and $\varepsilon_{c}$ presented a high correlation ($r_{Pearson}\approx$ 0.70) with TY (Table 3). Considering these results, the traits *A*, $\Delta_{leaf}$, NDVI${}_{slp}$ and tu, and the parameters $\varepsilon_{i}$, $\varepsilon_{c}$ and $\varepsilon_{p}$ were selected for a principal component analysis (PCA). The raw data from this section are available in the institutional dataset [29].

### 2.2. Characterization of the Potato Diversity Mini Core with Assessed Parameters

Some of the new adjusted values from the traits presented on Table 1 and assessed with a Pearson correlation, were considered in a PCA, which was performed to determine the main traits combination that promoted the ordination of accessions, to ascertain the main drivers that affected TY. The resulting first three components explained 76.5% of the total variance (Table 4). The coefficients having more weight on the calculation of the first component (PC1) were tu and NDVI${}_{slp}$, the latter with a negative sign. The parameters $\varepsilon_{c}$ and $\Delta_{leaf}$ presented high values (>0.70) and positive coefficients for the calculation of the second component (PC2). The third component (PC3), in turn, was negatively (−0.60) influenced by $\varepsilon_{i}$. Furthermore, a multiple regression (F value = 50.1, *p*-value < 0.001) was generated to calculate TY as a function of the three extracted PCs (R${}^{2}$ = 0.87), as follows:(1)TY=301.5+75.8×PC1+45.5×PC2+33.4×PC3

Wards’s method, applied for making a hierarchical clustering of the accessions assessed, showed three groups that minimized the total within-clusters variance (Figure 2A). The genotypes belonging to the three groups ordered in PC1 and PC2 space showed the following characteristics (Figure 2B): (i) the first one (GI) was characterized by late genotypes (highest tu values) with short senescence delay (lower NDVI${}_{slp}$), lower values of $\varepsilon_{i}$, $\varepsilon_{c}$ and high and low RUE, being scattered on the positive side of PC1. One third of the genotypes clustered into GI belonged to the *S. tuberosum subsp. andigenum*, whereas the second third to the species *curtilobum*, *juzepczuckii* and *stenotomum*, while the last third contained mainly assessed improved varieties (checks, see Section 4.2). (ii) The second group (GII), was formed by precocious genotypes with a long senescence delay (i.e., higher values of NDVI${}_{slp}$, and the lowest tu values), with high RUE, $\varepsilon_{i}$ and $\varepsilon_{c}$. GII was located on the negative and positive side of the PC1 and PC2, respectively and it was dominated by *S. tuberosum subsp. andigenum* (see Section 4.2). (ii) The third group (GIII), was characterized by precocious genotypes and with a long senescence delay (with high values of NDVI${}_{slp}$, and the lowest tu values) with high $\varepsilon_{i}$ and low RUE and $\varepsilon_{c}$, was spotted on the negative sides of the PC1 and PC2. GIII was conformed by species tuberosum (*subsp. andigenum* and *tuberosum*), *chauca*, *phureja* and *stenotomum* (see Section 4.2). The average ± standard error of TY for GI, GII and GIII were 226.7 ± 25.7, 511.2 ± 65.8 and 286.0 ± 42.7 g plant${}^{-1}$, respectively.

### 2.3. Efficiencies Diversity in Other Crops

It was used relative efficiencies—i.e., the ratio between experimental efficiencies and their respective theoretical maximum values as reported by the literature—for analyzing the current situation of potato crop yield improvement, compared to other crops produced worldwide. These relative efficiencies ($\varepsilon_{i}$, $\varepsilon_{c}$ and $\varepsilon_{p}$) were ranked and those calculated for potatoes in the study were compared to the ranking. The highest efficiencies obtained for potato genotypes in this study (Table 5) were: (i) $\varepsilon_{i}$ = 60.3% corresponding to 0.77 of the theoretical maximum $\varepsilon_{i}$ for major crops (90%—[4]); (ii) $\varepsilon_{c}$ = 6.4% corresponding to 0.68 of the theoretical maximum $\varepsilon_{c}$ for C3 plants (9.4%—[30]); (iii) $\varepsilon_{p}$ = 87%, near 0.97 of the highest theoretical $\varepsilon_{p}$ estimated for tuber crops (90%—[31]). Table 5, also contains the maximum efficiency values $\varepsilon_{i}$, $\varepsilon_{c}$ and $\varepsilon_{p}$ reported in the literature for C4 [32,33,34,35,36,37,38] and C3 [18,26,30,39,40,41,42,43,44,45,46,47,48,49,50] crops.

## 3. Discussion

### 3.1. Relationship among Efficiencies and Other Plant Traits

The highest TY values obtained were around those of native cultivars assessed by Tourneux et al. [52] (307–545.9 g plant${}^{-1}$), who considered a similar plant area (0.63 m${}^{2}$), and superior to the maximum value obtained by Condori et al. [7] (274.6 g plant${}^{-1}$), both assessed in the Bolivian Andes. These values were also superior to the maximum values of Kooman et al. [53] (229.5–366.8 g plant${}^{-1}$), who analyzed the variation in TY of eight cultivars among six sites from Europe and Africa. However, the cumulative intercepted PAR (PAR${}_{int})$ of its maximum yield cultivar (758 MJ m${}^{-2}$) was more than double of that intercepted by the highest yield (339.9 MJ m${}^{-2}$) in this experiment. These arguments supported our hypothesis and other studies [10,54] substantiating that potatoes can increase physiological efficiencies under diffuse radiation environments with a concurrent increase in TY.

The RUE values were within the wide range reported in the literature for *Solanum tuberosum*, based on PAR${}_{int}$ under field conditions (1.07–3.7 g MJ${}^{-1}$) [53,55,56]. According to Quiroz et al. [10], potatoes are able to show plasticity in RUE responses with an increment (5.4 g MJ${}^{-1}$, the highest value reported in the literature) under cloudy environments. Kooman et al. [53] also stated that largest variations in RUE are found at low radiation levels. In this study, the high RUE values obtained could be due to the low incident PAR during the trial (see Table 6), which could strongly influence the high yield responses in potato (see [10,54]). Even when only 25% of the cultivars got an $\varepsilon_{i}$ > 50%, this efficiency was an important component of GII and GIII, which clustered the genotypes with the highest yields (see Section 3.2). $\varepsilon_{i}$ was also the variable with the highest coefficient to calculate PC3 which explained 14.7% of the total cumulative variance (Table 4). Since PAR${}_{int}$ is determined by the aboveground biomass development, those accessions with reduced canopy cover and accelerated senescence rate presented low PAR${}_{int}$ values, and therefore, low $\varepsilon_{i}$ values. There was a good correlation between tu and $\varepsilon_{p}$ (r${}_{Pearson}$ = −0.62) in accordance with some studies [6,26,52], i.e., a low tu corresponded to a precocious earlier cultivar with high harvest index and also the best yields. Most of those with higher tu—late-maturing cultivar—developed more aboveground biomass than tubers, which gave them the lowest harvest index (HI) [57], and may appear to have low efficiencies due to greater respiration rate and physical loss of leaves than that of tubers [55]. The majority of the mini core representatives (50%) belong to sub-species *andigenum* (Table 7), cultivated in the Andes, where the minimum temperature (crucial for potato tuber induction [6,58]) is lower than the average temperature in the study site and thus, we hypothesized a delayed tuber initiation onset in those landraces. Tuber initiation in the present study (∼647–1089 °C-days, data not shown) was shorter than those reported by [7,59] using similar landraces (∼591–979 °C-days). In terms of calendar days tuber initiation delay is not apparent; 39–66 dap versus 44–79 dap reported by [7,59].

The most important correlation between TY and NDVI${}_{slp}$ was associated with the highest yields of those landraces which maintained their green foliage (longer senescence delay) longer, prior to senescence onset, i.e., they extended the duration of radiation interception thus increasing total yield [24,55,58]. Due to the close relationship between radiation interception and leaf area index with vegetation reflectance, the latter constitutes a reliable, quick and non-destructive measurement and thus suitable to be used in potato experiments [10,11,17,60]. High *A* and $\Delta_{leaf}$ values with not significant differences among accessions are evidences that genotypes were grown under non-limiting water and/or nutrient conditions [24,61,62]. This study reinforces the use of proximal/remote sensing as proxies for yield prediction of a diversity of potato genotypes.

### 3.2. Defining the Functional Diversity of Potato Mini Core

The interpretation of each retained component from the PCA, was based on the absolute coefficient value and direction (sign) for each trait included in the analysis (Table 4, see Section 2.2). Therefore, the most important variable, i.e., those with highest coefficients to calculate PC1, were NDVI${}_{slp}$ and tu as well as some radiation-related efficiencies. In other words, genotypes favoring tuberization precocity and senescence delay tend to present higher coefficients or weights on the principal component. PC1 presented the highest influence on TY (Equation (Equation 1)). Following the same logic, PC2 was related to the capability of plants to process the intercepted radiation; and PC3, in turn, was associated with the capability of plant to harvest incident radiation. TY for each accession and cluster can be estimated from Equation (Equation 1), using the coefficients of the principal components as shown in Table 4 and Figure 2. Average yield for GII is 2.3 times the average yield of GI, and 1.8 that of GIII. Depending on the quadrants the accessions of each group are located, the sign of the coefficients—dominated by the efficiencies and precocity—determines total yield. Moreover, TY was also positively correlated with $\varepsilon_{i}$ and $\varepsilon_{c}$ (Table 3), attesting that the combination of these efficiencies influence yield [4,12,30,53]. There was also a strong influence of early maturity, i.e., low tu–achieving maximum tuber growth early before other groups [26], and the delay of senescence—high NDVI${}_{slp}$—[21] on TY (see Equation (Equation 1)). Senescence delay allows higher accumulation of intercepted radiation and its transformation into biomass, thus explaining the reported positive relation between senescence duration and dry tuber yield [24,63]. Except for CIP 704393 (*S. stn. Goniocalyx*), the GII maintained *A* values between 23.0 and 28.6 μmol CO${}_{2}$ m${}^{-2}$ s${}^{-1}$ at 104 dap, corresponding to an optimum photosynthetic capacity [62], well related with senescence delay. In contrast, late maturity, i.e., high tu or the longer time required for tuber maximum growth rate, was associated to low yields in GI.

The low radiation conditions, typical of the study area (see Rinza et al. [54]), promoted a better response in terms of efficiency, by obtaining higher yield than experiments in the same study site [54,64], and those commonly obtained by local farmers in the Highlands [7]. Genotypes which could potentially delay their senescence constitute a crucial source of germplasm for the genetic improvement of important crops [65]. Condori et al. [7] stated that when cultivars were tested under different climates, all tested material responded in a similar manner, but the magnitude of the response varied depending on the cultivar. Most of these advantages were identified on landraces from the subspecies *andigenum*, which has a large range of geographic origin, large morphological variation and adaptation to harsh conditions, and thus, has been widely cultivated throughout the Andes [66], at altitudes between 1950–4500 m [67]. The very low tuber production of the diploid cultivar CIP 702650 (*S. ajanhuiri*) might be due to its endemic condition to the high Andean Altiplano—between southern Peru and central Bolivia—at elevations between 3700–4100 m a.s.l. [66]. In its habitat, this landrace is characterized by high RUE, HI and TY traits [7], showing the opposite response in the conditions of the present study.

### 3.3. Efficiencies in Relation to Other Crops

In the comparison of $\varepsilon_{i}$ among major crops according to FAO [68], potato was ranked in third place, after winter wheat (*Triticum aestivum*) and soybean (*Glycine max*), and above the responses of cassava (*Manihot esculenta*) and maize (*Zea mays*). In relation to $\varepsilon_{c}$, it was necessary to separate these major crops by crop type and to consider the energy content of biomass in seeds (23 MJ kg${}^{-1}$) and other organs (17 MJ kg${}^{-1}$) [51]. Potato is surpassed by rice (*Oriza sativa*) and winter wheat as C3 plants. Although C4 plants are more efficient in $\varepsilon_{c}$ due to the absence of photorespiration [30], when the comparison is based on the highest attained experimental value divided by their own theoretical maximum, the response for maize is similar to that of the potato. Concerning to $\varepsilon_{p}$, the theoretical maximum differed among types of crops, being the maximum for grains and seeds 65%, but around 90% for tuber and roots crops like potato, cassava, sweetpotato (*Ipomea batata*) and sugar beet (*Beta vulgaris*). In this ranking of relative efficiencies of most important crops, potato is above other root and tuber crops such as cassava [18] as well as other highly efficient crops such as the C3 crop soybean, that reached 92% of its own theoretical maximum. In addition, many of these crops have presented a delayed senescence under stress conditions [65].

## 4. Materials and Methods

### 4.1. Study Site and Crop Management

The study was conducted at the International Potato Center (CIP) experimental station (12.08° S, 76.95° W and 244 m a.s.l.) located in Lima, Peru from 5 July to 23 November 2017. The climate of this region is characterized by low rainfall (6.0 ± 0.74 mm of average yearly precipitation), high relative humidity and low vapor pressure deficit (81.2 ± 1.6% and 0.5 ± 0.1 kPa of average monthly values, respectively) (2013–2017, CIP’s Meteorological Station). During the experiment, the average monthly global radiation, minimum and maximum temperatures were 11.8 ± 2.8 MJ m${}^{-2}$, 12.7 °C (23 July) and 24.6 °C (24 October), respectively (see details in Table 6). The soil was a sandy–loam texture with average pH of 7.6 ± 0.01 and high, medium and low phosphoric (38.3 ± 0.6 ppm), potassium (187.7 ± 4.04 ppm) and organic matter content (1.35 ± 0.06%) respectively (Universidad Nacional Agraria La Molina, Soil, Plant, Water and Fertilizer Analysis Lab, Lima–Peru). Compound fertilizer was applied with an NPK dose of 180-100-160 kg ha${}^{-1}$ at planting for an effective area of 437.4 m${}^{2}$, where only 50% of N was applied and the remaining was incorporated at the second hilling (40 dap). Pest control included yellow traps and two chemical applications: at 63 dap, with Vertimec (Syngenta Crop Protection AG, Switzerland) and Sunfire (BASF SA, Brazil), and at 70 dap, with Movento (Bayer AG, Germany) and Trigard (Farmagro, Peru). The water was applied conventionally by furrow irrigation every nine days and the total quantity of water received by the crop was 4970 m${}^{3}$ ha${}^{-1}$.

### 4.2. Plant Material and Experimental Design

All plant materials for this study were acquired from the CIP genebank collection [27]. Twenty of the forty-five accessions from the potato mini core subset [28], belonging to seven species—including four subspecies—and four advanced or improved varieties, resistant to late blight and with highland adaptability—as checks—(Table 7), were planted in an augmented block design. Passport data and pictures for the mini core can be found at [28]. The accessions were randomly distributed in four blocks, each containing five genotypes and the four checks (improved varieties). Ten plants per accession were sown in a 13.5 m long single row (experimental unit) with distances between plants and rows of 1.5 and 1.8 m, respectively.

### 4.3. Data Acquisition

#### 4.3.1. Remote Sensing Imagery

Canopy cover (CC) and the normalized difference vegetation index (NDVI) were extracted from images recorded by an airborne platform consisting of an oktocopter (Mikrokopter, Germany) equipped with a multispectral camera (Tetracam Inc., USA). A total of 13 flights throughout the growing season at weekly intervals, with 120 m height above the ground level and 3 m s${}^{-1}$ speed during midday, were conducted. The images were acquired at 10 bits in 3 bands (Green, Red and Near Infrared) with 1.3 MPixels (1280 × 1024 pixels) spatial resolution. They were geometrically corrected using control points on the ground and libraries of the QGIS 3.8 platform (QGIS Development Team, 2019). The NDVI values were represented by the mean of each plot, whereas CC was computed following a segmentation process between soil and vegetation considering a threshold value on soil adjusted vegetation index (SAVI), for more details see Cucho-Padin et al. [69].

#### 4.3.2. Physiological Assessment

The net photosynthesis rate (*A*) was recorded on five occasions throughout the growing season (51, 64, 78, 91 and 104 dap) using a portable photosynthesis system (LI-6400XT model, Li-Cor Bioscience, Lincoln, NE, USA) monitoring three central target plants per accession, following Ramírez et al. [62] procedure. Additionally, the carbon isotopic discrimination in leaves ($\Delta_{leaf}$) was assessed by collecting leaf composed samples from each accession, twice (51 and 79 dap). The samples were prepared, and the results processed following the methodology of Ramírez et al. [61]. The photosynthetic performance of each accession was represented by an average value of *A* and $\Delta_{leaf}$ for the whole growing season.

Potato tubers were harvested at the end of the growing season, 121 and 141 dap, depending on each accession’s senescence stage code, according to the classification of Jefferies and Mackerron [70], i.e., only those with a majority of plants with codes 670 (yellowing of stems) and 690 (stems brown and fallen to the ground), were harvested (Table 7). Dry tuber yield per plant (TY) was determined by oven–drying tubers at 60 °C for 72 h and weighing.

### 4.4. Radiation-Related Parameters Calculation

In order to determine representative values of efficiencies per accession within the study, it was necessary to adjust the area occupied by a single plant, by averaging the maximum area achieved per genotype through the growing season, which resulted in 0.78 m${}^{2}$. In addition, the potato growth model SOLANUM, based on radiation interception and utilization (available on [71]), was employed to estimate total dry biomass (TB) and some key parameters (tu). This model—developed and validated by CIP with measured experimental data from a representative sample of the genetic diversity [7,59,72]—is described by eight growth parameters (see more details in [7]). Values of fresh and dry tuber data were used to ensure a good estimation (<5% of average relative error between observed and simulated data).

#### 4.4.1. Radiation Interception Efficiency ($\varepsilon_{i}$)

The cumulative incident PAR (PAR${}_{inc}$) was measured with a quantum sensor (LI190SB model, Li-Cor Bioscience, Lincoln, NE, USA) and recorded hourly with a data logger (CR1000 model, Campbell Sci. Inc., Logan, UT, USA) from 1 dap until harvest day. To estimate the intercepted PAR (PAR${}_{int}$) it was necessary to know the fraction of incident radiation intercepted by the plant canopy (*f*), which was estimated from daily CC data [53] fitted with the growth function Beta [73]. The PAR${}_{int}$ and $\varepsilon_{i}$ were calculated as follows:(2)$$PAR_{int}\left(MJ\;m^{-2}\right)=f\times PAR_{inc}$$
(3)$$\varepsilon_{i}\left(\%\right)=\frac{PAR_{int}}{PAR_{inc}}\times 100$$

#### 4.4.2. Conversion Efficiency ($\varepsilon_{c}$)

The conversion of radiation intercepted into biomass was determined by the ratio between the parameter radiation use efficiency (RUE) and the energy content on the plant mass, which was assumed to be 17.5 MJ kg${}^{-1}$ or 57.14 g MJ${}^{-1}$ [4]. Under non-limiting conditions, RUE is considered stable throughout the growing season and could be expressed as the ratio between total biomass (TB) produced, and PAR${}_{int}$ [74]. TB (in g plant${}^{-1})$ was estimated for each accession by using the SOLANUM model (see Section 4.4). RUE and $\varepsilon_{c}$ were calculated as follows:(4)$$RUE\left(g\;MJ^{-1}\right)=\frac{TB}{PAR_{int}}$$
(5)$$\varepsilon_{c}\left(\%\right)=\frac{RUE}{57.14}\times 100$$

#### 4.4.3. Partitioning Efficiency ($\varepsilon_{p}$)

The amount of the total biomass energy partitioned into the harvested portion of the crop [30] was described by the harvest index (HI), which was calculated as the ratio between TY and TB. $\varepsilon_{p}$ was calculated as follows:(6)$$\varepsilon_{p}\left(\%\right)=HI\times 100$$

### 4.5. Senescence Delay and Precocity Proxies

The slope of the decline of NDVI from its maximum value along the time, fitted by a linear function (NDVI${}_{slp}$), was used as indicator of senescence delay (higher values means longer senescence delay) [22,23]. The tuberization precocity proxy was represented by tu [26]—parameter from the Gompertz function, modeled by SOLANUM that indicates the time when the maximum tuber partitioning rate occurred (see Condori et al. [7] for further details). Higher values in tu means lower tuberization precocity i.e., late-maturing cultivar.

### 4.6. Data Analysis

TY, physiological (*A* and $\Delta_{leaf}$), radiation-related ($\varepsilon_{i}$, $\varepsilon_{c}$ and $\varepsilon_{p}$), NDVI${}_{slp}$ and tu parameters were adjusted to an analysis of variance for augmented block design (named DAU analysis) and a Tukey test, for significant differences between accessions (both analyses made with the “agricolae” package [75]). The new adjusted values were assessed with a Pearson correlation and some of them, were considered in a principal components analysis (PCA), to ascertain the main drivers that affected TY. To group the genotypes based on the traits used in PCA a Wards’ method for hierarchical clustering was performed using the R package “FactoMineR” [76]. Also, a multiple linear regression analysis was performed between the scores of the main principal components and TY. All the analyses were run by R software [77]. Databases such as Faostat [68], Scopus [78] and Google Scholar [79] were consulted for analyzing efficiency of potato in relation to other major crops.

## 5. Conclusions

There are still options for improving $\varepsilon_{i}$ and $\varepsilon_{c}$ in potato, which are quite far from their theoretical maximum, both efficiencies were important in the ordination of the germplasm diversity and yield in the assessed mini core potato panel. Senescence delay in combination with low tuberization precocity are traits apparently related to a major option for a longer carbon fixation, but at the same time more possibility to allocate this fixed carbon to tuber productions. More physiological studies are recommended to confirm the usefulness of these traits, and thus also the identification of molecular genetic markers for their assessment in breeding programs. It is also necessary to assess these genotypes in other environments for a better characterization of their physiological and agronomic performance. The use of remote sensing tools for a rapid and non-destructive assessment of canopy reflectance decay, as a surrogate of senescence delay (closely related to tuber yield), are promissory in the assessment of field high-throughput phenotyping platforms. The determination of coefficients related to tuber yield and senescence delay (through remote sensing inspection) under different contrasted environments, could help in the targeting of the population of environments for worldwide potato breeding programs.

## Figures and Tables

**Figure 1 plants-09-00787-f001:**
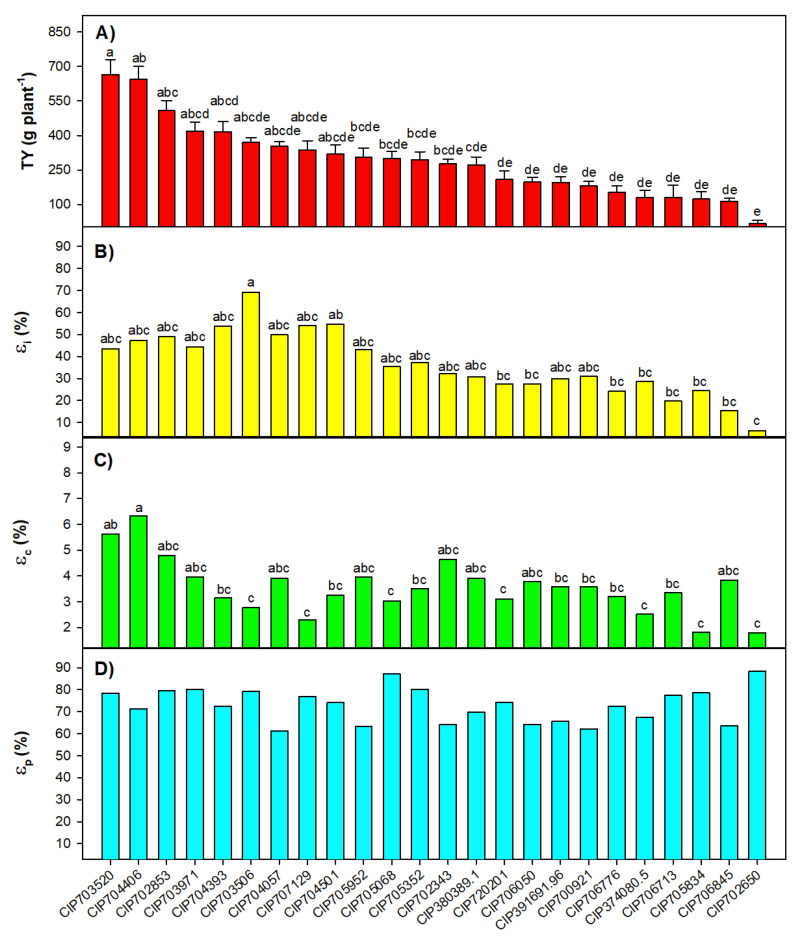
Adjusted mean values of TY—dry tuber yield per plant (**A**); $\varepsilon_{i}$—radiation interception (**B**), $\varepsilon_{c}$—conversion (**C**) and $\varepsilon_{p}$—partitioning efficiency (**D**) per accession. Different letters mean significant differences (*p*-value < 0.05) among them, according to a Tukey test. There were no significant differences (*p*-value > 0.05) for $\varepsilon_{p}$.

**Figure 2 plants-09-00787-f002:**
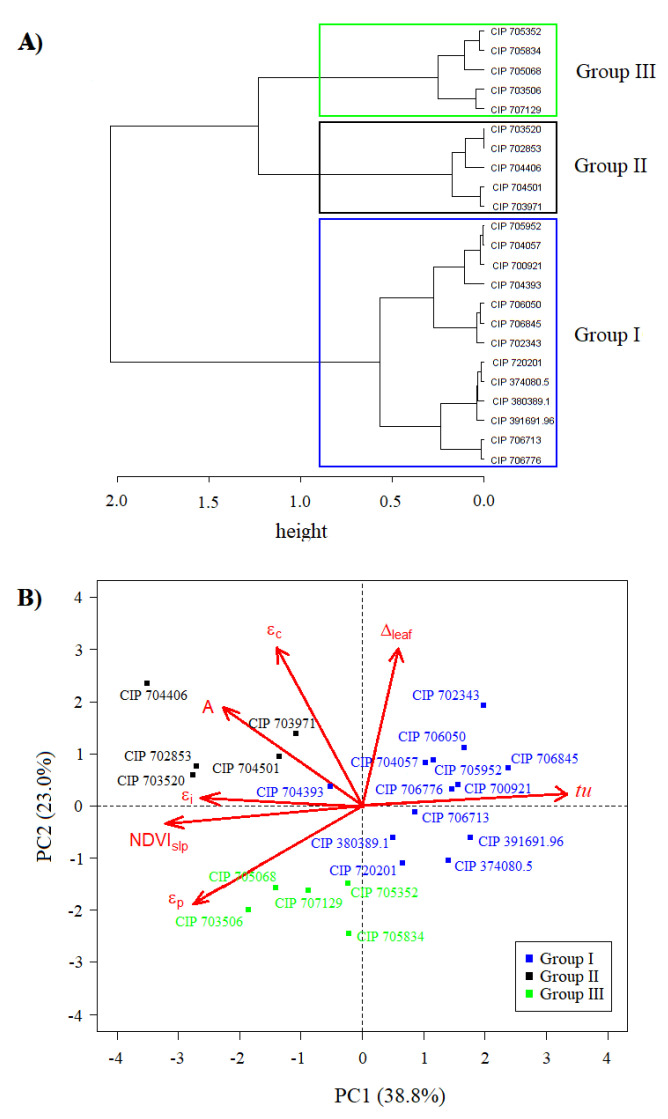
Hierarchical clustering using Ward’s method (**A**). Accessions ordination based on principal component analysis for physiological (*A*, $\Delta_{leaf}$), radiation-related ($\varepsilon_{i}$, $\varepsilon_{c}$, $\varepsilon_{p}$), senescence delay ($NDVI_{slp}$) and tuberization precocity (tu) parameters (**B**). See abbreviations in Table 3.

**Table 1 plants-09-00787-t001:** F-values corresponding to the analysis of variance for an augmented block design.

Trait	Check	Augmented	Check × Augmented
TY (g plant${}^{-1}$)	5.24 *	11.7 ***	38.1 ***
$\varepsilon_{i}$ (%)	0.17 n.s.	4.62 *	14.5 **
$\varepsilon_{c}$ (%)	7.80 **	5.78 **	5.01 n.s.
$\varepsilon_{p}$ (%)	1.53 n.s.	1.35 n.s.	3.78 n.s.
*A* (μmol CO${}_{2}$ m${}^{-2}$ s${}^{-1}$)	3.20 n.s.	4.9 **	15.3 **
$\Delta_{leaf}$ (‰)	0.88 n.s.	2.02 n.s.	15.8 **
NDVI${}_{slp}$	1.74 n.s.	2.31 n.s.	3.04 n.s.
tu (°C d)	0.30 n.s.	1.41 n.s.	3.5 n.s.

n.s., *, ** and *** means *p*-value > 0.05 (not significant), *p*-value < 0.05, *p*-value < 0.01 and *p*-value < 0.001 respectively. TY (g plant${}^{-1}$)—Dry tuber yield per plant; $\varepsilon_{i}$ (%)—Interception efficiency; $\varepsilon_{c}$ (%)—Conversion efficiency; $\varepsilon_{p}$ (%)–Partitioning efficiency; *A*—Net photosynthesis rate; $\Delta_{leaf}$—Carbon isotopic discrimination in leaves, NDVI${}_{slp}$—Senescence delay; tu–Tuberization precocity.

**Table 2 plants-09-00787-t002:** Adjusted values of physiological (*A*, $\Delta_{leaf}$), senescence delay (NDVI${}_{slp}$) and tuberization precocity (tu) parameters for each accession.

Accession Number	*A* (μmol CO${}_{2}$ m${}^{-1}$ s${}^{-1}$)	$\Delta_{leaf}$ (‰)	NDVI${}_{slp}$ ($\times 10^{2}$)	tu (°C day${}^{-1}$)
CIP 700921	20.4 ± 2.36	22 ± 0.4	−0.41	531
CIP 702343	21.0 ± 2.08	23 ± 0.2	−0.51	556
CIP 702650	-	-	−0.50	376
CIP 702853	24.7 ± 1.14	22 ± 0.2	−0.31	391
CIP 703506	19.1 ± 1.58	21 ± 0.0	−0.31	431
CIP 703520	24.9 ± 0.82	21 ± 0.1	−0.31	396
CIP 703971	26.2 ± 1.24	23 ± 0.6	−0.44	476
CIP 704057	21.3 ± 1.94	22 ± 0.4	−0.44	556
CIP 704393	20.7 ± 2.25	23 ± 0.0	−0.34	476
CIP 704406	26.6 ± 0.58	22 ± 0.1	−0.24	356
CIP 704501	24.4 ± 1.83	23 ± 0.3	−0.44	396
CIP 705068	23.6 ± 1.56	21 ± 0.2	−0.44	416
CIP 705352	19.5 ± 2.28	21 ± 0.5	−0.41	471
CIP 705834	20.3 ± 1.83	21 ± 0.1	−0.34	436
CIP 705952	21.7 ± 1.79	22 ± 0.2	−0.41	556
CIP 706050	24.3 ± 1.85	22 ± 0.5	−0.51	551
CIP 706713	22.8 ± 1.31	22 ± 0.4	−0.54	476
CIP 706776	24.1 ± 2.48	22 ± 0.3	−0.61	496
CIP 706845	17.0 ± 1.59	23 ± 0.4	−0.51	456
CIP 707129	22.4 ± 2.02	21 ± 0.2	−0.34	496
CIP 374080.5	21.7 ± 2.30	21 ± 0.1	−0.50	495
CIP 380389.1	20.3 ± 2.20	21 ± 0.0	−0.40	490
CIP 391691.96	19.5 ± 1.50	21 ± 0.6	−0.50	520
CIP 720201	21.7 ± 1.60	21 ± 0.4	−0.45	500

See abbreviations in Table 1.

**Table 3 plants-09-00787-t003:** Pearson correlation matrix among dry tuber yield per plant (TY), physiological (*A*, $\Delta_{leaf}$), radiation-related ($\varepsilon_{i}$, $\varepsilon_{c}$, $\varepsilon_{p}$), senescence delay (NDVI${}_{slp}$) and tuberization precocity (tu) parameters. In gray: correlations > |0.5| between variables.

	TY						
$\varepsilon_{i}$	0.69	$\varepsilon_{i}$					
$\varepsilon_{c}$	0.70	0.12	$\varepsilon_{c}$				
$\varepsilon_{p}$	0.27	0.24	−0.21	$\varepsilon_{p}$			
A	0.55	0.23	0.44	0.27	A		
$\Delta_{\mathrm{leaf}}$	0.06	0.04	0.29	−0.46	0.16	$\Delta_{\mathrm{leaf}}$	
NDVI${}_{\mathrm{slp}}$	0.74	0.69	0.25	0.38	0.21	−0.13	NDVI${}_{\mathrm{slp}}$
tu	−0.54	−0.03	−0.26	−0.62	−0.37	0.08	−0.59

See abbreviations in Table 1.

**Table 4 plants-09-00787-t004:** Loadings of the first three principal components (PC) resulting from the ordination of physiological (*A*, $\Delta_{leaf}$), radiation-related ($\varepsilon_{i}$, $\varepsilon_{c}$, $\varepsilon_{p}$), senescence delay (NDVI${}_{slp}$) and tuberization precocity (tu) parameters. In gray: scores > |0.5|. TCV = total cumulative variance.

Trait	PC1	PC2	PC3
$\varepsilon_{i}$	−0.66	0.04	−0.60
$\varepsilon_{c}$	−0.35	0.77	0.00
$\varepsilon_{p}$	−0.69	−0.46	0.42
*A*	−0.56	0.46	0.47
$\Delta_{leaf}$	0.15	0.76	−0.03
NDVI${}_{slp}$	−0.80	−0.09	−0.47
tu	0.83	0.06	−0.25
Eigenvalue	2.72	1.61	1.03
TCV (%)	38.8	61.8	76.5

See abbreviations in Table 1.

**Table 5 plants-09-00787-t005:** Maximum values of efficiencies reported in the literature, for worldwide major crops according to FAO. Ref = References.

Plant Type	Crop Type	Crop	$\varepsilon_{i}$	Ref	$\varepsilon_{c}$	Ref	$\varepsilon_{p}$	Ref
	Perennial grass	Sugarcane	95.0	[32]	8.0	[33]	81.2	[34]
**C4**	Grain	Maize	55.0	[35]	8.7	[36]	47.1	[35]
	Grain	Sorghum	50.1	[37]	8.0	[38]	20.0	[37]
	Grain	Winter wheat	97.8	[39]	7.5	[40]	44.0	[41]
	Grain	Rice	80.0	[43]	6.6	[42]	62.0	[38]
	Leguminous	Soybean	89.0	[30]	4.3	[44]	60.0	[30]
**C3**	Tuber root	Sugar beet	90.0	[45]	6.2	[46]	86.0	[47]
	Tuber root	Cassava	64.0	[48]	1.4	[18]	70.5	[18]
	Tuber root	Sweetpotato	91.0	[49]	3.4	[26]	46.0	[50]
	Tuber	Potato	69.3 ${}^{†}$		6.4 ${}^{†}$		87.0 ${}^{†}$	

$\varepsilon_{i}$–theoretical maximum = 90% [4]; $\varepsilon_{c}$ (C4)–theoretical maximum = 12.3% [30]; $\varepsilon_{c}$ (C3)–theoretical maximum = 9.4% [30]; $\varepsilon_{p}$ (grain and seed crops)–theoretical maximum = 65% [30]; $\varepsilon_{p}$ (tuber crops)–theoretical maximum = 90% [31]; Energy content of the seeds = 23 MJ kg${}^{-1}$ [51]; Energy content of the biomass = 17.5 MJ kg${}^{-1}$ [4]; ${}^{†}$ Maximum values obtained in this study.

**Table 6 plants-09-00787-t006:** Average daily values (±SE) of environmental conditions during the experimental growing season 2017. PAR—Photosynthetically active radiation. VPD—Vapor pressure deficit.

Meteorological Variables	July	August	September	October	November
Maximum temperature (°C)	19.8 ± 0.29	19.2 ± 0.27	19.0 ± 0.38	21.5 ± 0.30	22.1 ± 0.26
Minimum temperature (°C)	15.3 ± 0.14	14.3 ± 0.10	14.0 ± 0.08	14.6 ± 0.08	15.2 ± 0.19
Average temperature (°C)	16.8 ± 0.10	15.8 ± 0.10	15.6 ± 0.12	16.8 ± 0.15	17.7 ± 0.18
Average relative humidity (%)	85.4 ± 0.61	88.7 ± 0.57	91.2 ± 0.59	89.6 ± 0.46	86.2 ± 0.57
Solar radiation (MJ m${}^{-2}$ day${}^{-1}$)	8.5 ± 0.77	8.8 ± 0.69	9.4 ± 0.96	16.8 ± 0.68	16.1 ± 0.73
PAR (MJ m${}^{-2}$ day${}^{-1}$)	3.7 ± 0.27	3.6 ± 0.27	3.8 ± 0.36	7.2 ± 0.27	6.8 ± 0.29
Average VPD (kPa)	0.24 ± 0.01	0.18 ± 0.01	0.14 ± 0.01	0.17 ± 0.01	0.24 ± 0.01
Maximum VPD (kPa)	0.68 ± 0.03	0.60 ± 0.03	0.56 ± 0.05	0.84 ± 0.03	0.92 ± 0.03

**Table 7 plants-09-00787-t007:** Selected accessions from the potato mini core subset (CIP Genebank) and improved varieties (checks) used in this study, with origin and harvest time for each of them. Groups were defined based on Ward’s method (see Figure 2). dap—days after planting.

Accession Number	Species	Ploidy	DOI	Contry of Origin	Harvest Time (dap)	Groups
CIP 700921	*Solanum tuberosum* *subsp. andigenum*	4×	10.18730/91RP	Peru	121	I
CIP 702343	*Solanum tuberosum* *subsp. andigenum*	4×	10.18730/9CJ=	Peru	121	I
CIP 702650	*Solanum ajanhuiri*	2×	10.18730/9EY0	Bolivia	121	-
CIP 702853	*Solanum tuberosum* *subsp. andigenum*	4×	10.18730/9GB8	Peru	141	II
CIP 703506	*Solanum phureja*	2×	10.18730/9R4=	Colombia	121	III
CIP 703520	*Solanum tuberosum* *subsp. andigenum*	4×	10.18730/9RHB	Colombia	141	II
CIP 703971	*Solanum tuberosum* *subsp. andigenum*	4×	10.18730/A4X0	Peru	141	II
CIP 704057	*Solanum tuberosum* *subsp. andigenum*	4×	10.18730/A7G9	Ecuador	121	I
CIP 704393	*Solanum stenotomum* *subsp. goniocalyx*	2×	10.18730/AGC$	Peru	141	I
CIP 704406	*Solanum tuberosum* *subsp. andigenum*	4×	10.18730/AGR9	Peru	141	II
CIP 704501	*Solanum tuberosum* *subsp. andigenum*	4×	10.18730/AKKT	Peru	121	II
CIP 705068	*Solanum tuberosum* *subsp. tuberosum*	4×	10.18730/B40∼	Chile	141	III
CIP 705352	*Solanum tuberosum* *subsp. andigenum*	4×	10.18730/BC0Y	Ecuador	141	III
CIP 705834	*Solanum stenotomum* *subsp. stenotomum*	2×	10.18730/BTDA	Peru	141	III
CIP 705952	*Solanum stenotomum* *subsp. stenotomum*	2×	10.18730/BXK1	Bolivia	121	I
CIP 706050	*Solanum juzepczukii*	3×	10.18730/C09D	Peru	141	I
CIP 706713	*Solanum tuberosum* *subsp. andigenum*	4×	10.18730/CJ0S	Argentina	121	I
CIP 706776	*Solanum curtilobum*	5×	10.18730/CKS8	Bolivia	121	I
CIP 706845	*Solanum stenotomum* *subsp. stenotomum*	2×	10.18730/CNTU	Bolivia	121	I
CIP 707129	*Solanum chaucha*	3×	10.18730/CS5*	Peru	141	III
**Checks**					
CIP 374080.5	Variety Perricholi	4×	10.18730/2BRK	Peru	121	I
CIP 380389.1	Variety Canchan	4×	10.18730/P5MJZ	Peru	141	I
CIP 391691.96	Variety Serranita	4×	10.18730/P5P8B	Peru	121	I
CIP 720201	Variety Yungay	4×	10.18730/D72∼	Peru	141	I

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
