# Peer review of "Radiation Interception, Conversion and Partitioning Efficiency in Potato Landraces: How Far Are We from the Optimum?"

_plants, 2020, doi:10.3390/plants9060787_

Round 1

Reviewer 1 Report

Plants Manuscript #: plants-777269

Authors: Cecilia Silva-Diaz et al.

Title: Radiation interception, conversion and partitioning efficiency in potato landraces: How far are we from the optimum

The authors have studied 20 landraces of potatoes having a diverse range of genotypes for many different physiologic, morphological and agronomical traits to identify traits and methodologies to optimize tuber biomass yield. They go on to compare key physiological and agronomical traits between potatoes and other tuber or root crops as well as seed crops, both C3 and C4 plants, with the goal of assesse what traits with higher capacity for crop improvement. The general strategy and goal are reasonable and of general interest.

Concerns / Questions about Scientific Conclusions/Statements:

The authors generally use sound data and methodologies to assess the range of aspects of potato growth and physiology. However, for some calculations and estimates there seem to be key assumptions made, some of which are not always clear there is sufficient evidence/reasoning to support. Further, for some of conclusion made when comparing potato to other crop plants, such as C4 seed-based crop plants, there is a logical “stretch” as to whether the comparisons are fully accurate in the absolute. For example, are comparing the absolute values of Efficiency of Partitioning (ep) or Efficiency of Conversion into Biomass (ec) between a tuber or root crop versus seeds valid and realistic? Also, the fact that all landraces were grown in the same environment (location, weather, etc…) has both advantages and disadvantages. It allows for comparisons between landraces, but it means important naturally occurring environmental factors (rain, temperature, soil nutrition) is taken out, which might alter some of the physiological traits and thus might affect conclusions.

Between Table 5, with various crop pants, there were major inconsistencies between data shown in table and what is written in the text, particularly Section 3.3 (Lines 204 – 215). One example, the text Lines 212-214 regarding Efficiency of Partitioning (ep) states that “potato occupies the first place and like cassava, ep was close to reaching its theoretical maximum, as well as the C4 crop soybean that reached 92% of its own theoretical maximum.” Problems in basic plant biology and in comparison to Table 5: first Soybean is not a C4 plant and second, cassava (60.2) is fourth, much lower than others in that list so does not seem near theoretical maximum. This section needs to be reviewed and rewritten to be consistent and accurate.

Further, I was genuinely uncertain at times when the authors were using Dry Tuber Yield (TY) and Total Plant Biomass (TB ??). I did not see a clear description of the method that was used to determine Total Plant Biomass (“total biomass”? TB) in the Methods section. Lines 290-291 suggest TB was determined from simulated harvest index with SOLANUM for each accession. Thus, it seems it was predicted from the Dry Tuber Weight and not directly measured from the actual plants. If so, please explain the assumptions/parameters, for that seems critical for determining the Harvest Index (HI), and my concern is that if it s simulated prediction the HI might not be as accurate as needed.

The citations and Reference section need significant edits/corrections. Although all of the in-text citations are present in the Reference section, many of the citations in the Reference are not in the correct order / numbering compared to the chronological order they appear in the text. For example, citation #22 appears in the text before citations #20 and #2; #24 appears before #23, #34 and #35 before #32 and #33; #44 appears before #42 and #43; and #48 appears before #47. Finally, the biggest issue with References, citations #57 - #72 are in the Reference section but are not in the text, anywhere. These order and References need to be correct.

Other scientific / substantive issues are smaller, but still need to be corrected or more explanation provided to clarify. These include:

L21-24: The authors should at least acknowledge that genetic engineering/editing can produce additional genetic diversity outside of the naturally occurring germplasm / genotypes.

L78-79: The statement, “About 50% of mini core accessions used in this study had a TY superior to the mean ...”. By definition, 50% of the values for any data set would be superior (above) the mean. That is the nature of the average.

L81-83: It is not completely clear why accession CIP 702650 (S. ajuanhuiri) was removed. Just because it was the poorest performer in some of the parameters, does not by itself justify its removal. Once it was removed, then there is a new low for the “above ground biomass”.

L101-102. Start of Section 2.2 needs a clear introductory sentence to the PCA.

L127-128 and Table 5 footnotes. The theoretical maximum value for Efficiency of Partitioning (ep) in tuber/root crops need to be stated in table footnote and the text. Also, all of these in Table 5 footnote need citations provided.

L142: For the “…RUE responses with important increments under …”, the meaning for the wording “important increments is not clear. Please reword for clarification.

L149-151: States that the, “…Efficiency of Radiation Interception (ei) values obtained were lower than expected and probably due to an overestimation of PARinc caused by clustering landraces in two harvesting groups …” This does not make sense, as I read it, for from the Methods the PARinc was determined by measuring for each landrace (Methods, 4.4.1). Please clarify.

L189: Define full name for TIO. This is the first and only use of this term.

L178: Mentions that “Higher weights tend to favor tuberization …” Not clear what weights are being referred to here. TY, TB, both, something else? Please clarify.

L179-181: States, “It seems that the significant differences in yield, defined by the intrinsic properties of each genotype [3], were principally determined by the combination of ei and ec [3,11,28,30].”  I don’t see this from either Table 3 (Pearson Correlations) PCA analysis (PC2 specifically) from Table 4. Please clarify.

L198: CIP 702650 is diploid, and has low tuber production. It seems it would be worth testing correlation between tuber production (TY) and general ploidy across the 20 landraces used here. It seems likely these are known.

L201: Define full name for DTY. This is the first and only use of this term.

L204-205: Potato being third rank for eI from Table 4 is not so dramatic or meaningful considering there are only a total of 6 places with date (missing data from four of the crop plants). That puts potato right in the middle (50%), so not dramatic.

L207-208: Regarding ec, it states that, “Potato is surpassed by rice (Oriza sativa) and winter wheat as C3 plants.” This is not correct based on data in Table 5. In fact among C3 plants rice is 4th position and below (not above) potato. Please correct to reflect evidence.

L212-214: As mentioned above, there are numerous errors/inconsistencies with this sentence/section. Regarding Efficiency of Partitioning (ep), section states that, “…potato occupies the first place and like cassava, ep was close to reaching its theoretical maximum, as well as the C4 crop soybean that reached 92% of its own theoretical maximum.” First problem, Soybean is not a C4 plant. Second, cassava (60.2) is fourth, much lower than others in that list so does not seem near theoretical maximum.

L257: Please define A trait, net photosynthesis rate (okay to duplicate what is mentioned in Results section, page 3).

L267-268: Make clear in Methods that Tuber dry-mass is the TY. Further, it mentions samples were weighed dries sub-samples. Then, oven-drying at 60oC for 72 h and reweighing. Not clear, were they dried twice?

L288-289: States, “Under non-limiting conditions, REU is considered stable throughout the growing season ..”   Do you know these were all grown under non-limiting conditions with respect to nutrients, water, light, etc…?

L290-291: It is not clear to me exactly how the total plant biomass (TB) was calculated. It seems TB was estimated based on the harvest index through a simulation using SOLANUM. Thus, the actual TB was determined. This seems like a potential problem if the simulation is not accurate for each landrace, and not citation was provided. Especially since the TB is used to calculate the harvest index. There seems to be a circular argument here. Harvest Index used to estimate TB, then (Line 294) TB used for the calculation of Harvest Index. Please clarify.

L304: Define full name for ABD, analysis of variance model for augmented block design.

The rest of my concerns and suggestions focus on details in the writing and some specific grammatical suggestions/issues (the ones that I found in my reading):

L47: Change “along” to “throughout” its growing period.   Also, change “is” to “are” so it reads,
“…sensed data are contingent …”

L65: The word “subsets” should be plural.

Table 1: First Column heading of “Hi” is not clear. Seems like this should be “Trait”. Clarify.

Table 1 Footnote: should read, “not significant”

Table 3 Row heading for Efficiency of Conversion (ec), should be Efficiency of Partitioning (ep).

Figure 1: I would move the Panel labels “A”, “B”, “C”, and “D” to the left side of graphs. These were easy to miss on the right side.

L108: Change “which” to “that”, so that it reads, “..three groups that minimized …”

Table 5 footnotes: Use “;” to separate different types of efficiencies. This needs citations and the maximum theoretical efficiency of Efficiency of Partitioning ep for tuber and root crops.

L204: Change the word “for” to “of” so that it reads, “…comparison of ei among major crops…”

L277: Change “was” to “were” so it reads, “…tuber data were used …”

Reviewer 2 Report

I have two relatively major comments: the first one is related to the choice of material. How the panel were chosen from the minicore collections is not clear. This panel doesn’t appear to be very diverse for traits under consideration. In addition, genetic variation for 'partitioning efficiency'-one key trait that is subject to investigation- is lacking. How could this could have impacted the overall results since this is part of the yield potential equation? The other concern is related to the fact that it was conducted for only one field season and at one location while most of traits under investigated are subject to important 'genotype x environment' interactions. This should be reflected in the interpretation of the result and conclusion. It would be good also to have significance level of the Pearson correlation coefficients shown. Details on material preparation could also be useful. Otherwise the manuscript is very well written!

Minor comments:

  • Line 62: .. the International potato centre (CIP) preserves and custody the most important…. is the word 'custody' used here as a verb?
  • Line 131: the higher TY values obtained were … did you mean the highest?

134: 'It' was also…. It is not clear what 'It' refers to.
